# Sustainability of C-Reactive Protein Apheresis in Acute Myocardial Infarction—Results from a Supplementary Data Analysis of the Exploratory C-Reactive Protein in Acute Myocardial Infarction-1 Study

**DOI:** 10.3390/jcm11216446

**Published:** 2022-10-31

**Authors:** Horst Skarabis, Jan Torzewski, Wolfgang Ries, Franz Heigl, Christoph D. Garlichs, Rudolf Kunze, Ahmed Sheriff

**Affiliations:** 1Statistical Consultant, Groß-Oesingen, 29393 Gifhorn, Germany; 2Cardiovascular Center Oberallgäu-Kempten, 87439 Kempten, Germany; 3Medical Clinic, Diakonissenhospital Flensburg, 24939 Flensburg, Germany; 4Medical Care Center Kempten-Allgäu, 87439 Kempten, Germany; 5Pentracor GmbH, 16761 Hennigsdorf, Germany; 6Department of Gastroenterology, Infectiology and Rheumatology, Charité University Medicine Berlin, 10117 Berlin, Germany

**Keywords:** C-reactive protein, STEMI, AMI, CRP apheresis, CMR, MRI, infarct size, inflammation, inflammatory mediators, ischaemia, hypoxia, phagocytosis

## Abstract

In the multicenter, non-randomized, exploratory **C**-reactive protein (CRP) **A**pheresis in **M**yocardial **I**nfarction (**CAMI**-1) study, CRP apheresis after ST-Elevation Myocardial Infarction (STEMI) significantly decreased blood CRP concentrations in humans. Cardiac damage was assessed by Cardiac Magnetic Resonance (CMR1) 3–9 d after onset of STEMI symptoms and quantified by myocardial infarct size (IS; %), left ventricular ejection fraction (LVEF; %), circumferential strain (CS) and longitudinal strain (LS). Compared with the control group (*n* = 34), cardiac damage was significantly lower in the apheresis group (*n* = 32). These findings suggested improved wound healing due to CRP apheresis already within few days after the STEMI event. In the current supplementary data analysis of CAMI-1, we have tested by a follow-up CMR (CMR2) after an average of 88 (65–177) d whether the effect of CRP apheresis is clinically maintained. After this time period, wound healing in STEMI is considered complete. Whereas patients with low CRP production and a CRP gradient cut off of <0.6 mg/L/h in the hours after STEMI (9 of 32 patients in the CRP apheresis group) did not significantly benefit from CRP apheresis in CMR2, patients with high CRP production and a CRP gradient cut off of >0.6 mg/L/h (23 of 32 patients in the CRP apheresis group) showed significant treatment benefit. In the latter patients, CMR2 revealed a lower IS (−5.4%; *p* = 0.05), a better LVEF (+6.4%; *p* = 0.03), and an improved CS (−6.1%; *p* = 0.005). No significant improvement, however, was observed for LS (−2.9%; *p* = 0.1). These data suggest a sustained positive effect of CRP apheresis on heart physiology in STEMI patients with high CRP production well beyond the period of its application. The data demonstrate the sustainability of the CRP removal from plasma which is associated with less scar tissue.

## 1. Introduction

In case of acute, aseptic tissue damage such as STEMI, CRP concentration in the blood, within few hours, can increase up to 100–200 times over the initial value [1,2]. Whereas a massive increase in CRP concentration may be useful as a first defense against microbial infections, it is likely counterproductive in injuries such as coronary occlusions. Indeed, the extent of myocardial damage after STEMI correlates significantly with the velocity of CRP increase during the first 1–3 d [1,3]. Obviously, the latter findings are in line with the results of the exploratory CAMI-1 study [4]. The amount of CRP, expressed as its increase within the first 32 h after STEMI, correlates with the extent of myocardial damage (% IS), functional impairment (% LVEF) and strains (% LS, CS). Notably, in the latter study, CRP removal from the human plasma by specific CRP apheresis in the two to three days after STEMI reduced heart damage as assessed by CMR (CMR1) [4]. Therefore, CAMI-1 has successfully confirmed the efficacy and feasibility of CRP apheresis.

Two issues are highly relevant for patient prognosis after STEMI: First, a timely revascularization within 2 h after the onset of STEMI symptoms. The latter will result in almost complete preservation of cardiac function with a very good long-term prognosis. Secondly, and mainly in case of delayed intervention, the healing process during the weeks after STEMI. This healing process of injured myocardial tissue is supported by medical therapeutic management in the ICU.

Up to the present day, a “healing process” has been discussed for the myocardium after STEMI [5]. The concept of a “healing process”, however, is misleading. In fact, a STEMI primarily causes only undersupplied tissue (area at risk), which switches its metabolism to “anaerobic”. The resulting damage is induced mainly by the lack of energy (less adenosine triphosphate (ATP) per glucose molecule by factor 16) [6,7,8]. This indeed causes stunning, but not necessarily immediate cell death. The ATP deficiency, however, subsequently causes neglect of membrane maintenance. The outer cell membrane now exposes lipids, especially lyso-phosphatidylcholine, which indicate apoptosis. CRP then binds to the membrane, which in turn induces the binding of complement. The immune system immediately disposes of apoptotic cells, and this is obviously a major pathway of tissue destruction after STEMI. The process is completed after about three months [9,10,11].

CRP apheresis is obviously important for STEMI patients with delayed revascularization. CRP apheresis achieves the local absence of CRP in the myocardium. It thereby provides more time to the cells to switch back from anaerobic to aerobic metabolism in order to be able to revert the apoptotic appearance of their outer cell membranes to prevent disposal by phagocytes.

Whereas CMR1 2–9 d after STEMI was subject matter of the original CAMI-1 publication, we have now taken a closer look at the data of the second CMR (CMR2) and whether the improvement of cardiac function observed in CMR1 is sustained for an average of 88 (65–177) d after the STEMI event. This report thus investigates the long-term sustainability of the initial beneficial effect of CRP apheresis after STEMI with respect to the CMR outcome parameters IS, LVEF, LS and CS.

## 2. Materials and Methods

The basis for this secondary analysis was the data generated in CAMI-1 [4]. In this prospective, controlled, multi-center, non-randomized, exploratory pilot study, the safety and efficacy of CRP apheresis were examined. Furthermore, the association of CRP levels or CRP gradients with myocardial infarct size and cardiac function was studied. Study design, patient population, trial protocol, CRP quantification, CRP apheresis, CMR and image acquisition as well as image analysis have been described in detail [4]. Inclusion and exclusion criteria are also described in detail [4]. In addition, 83 patients with acute myocardial infarction (STEMI) were recruited. Forty-five of them received CRP apheresis, and 38 patients comprised the control group. Thirty-two patients from the apheresis group and 34 from the control group were finally included in the statistical analysis. The increase rate of the CRP concentration (defined as gradient = CRP_grad_) within the first 32 h after onset of symptoms was used as the basis for analysis. Myocardial parameters were measured using CMR. The main cardiac parameters were % IS, % LVEF, % LS and % CS. CMR1 was assessed 2–9 d after onset of STEMI symptoms, and CMR2 was assessed 88 (65–177) d after STEMI. The baseline characteristics of both groups are comparable [4]. The only two variables which were significantly lower in the control group were BMI (*p* = 0.05) and diabetes (*p* = 0.03).

### 2.1. Statistical Analysis

In contrast to randomized studies, observational studies require an adjustment regarding possible confounders to prove a so-called “cause-and-effect relationship”. This adjustment was done here in the same way as in the original CAMI-1 publication by including the estimated propensity based on age, sex, body mass index (BMI), hypertension, etc. in statistical modeling [4]. The CRP-Gradient, defined as the increase rate of CRP within the first 32 h after STEMI onset with a cut-off = 0.6 mg/L/h, served as the decisive key for statistical analysis. The statistical analyses were performed by using the software R, version 3.6.1.

### 2.2. Study Approval

The CAMI-1 study was approved by the Ethics Committee No.: 042/15 (I), Medical Association Schleswig-Holstein, Germany. The study was registered under the number WHO ICTRP: DRKS00008988. Written informed consent was received from all participants prior to inclusion in the study.

## 3. Results

### 3.1. Comparison of CMR1 and CMR2

Table 1 and Table 2 compare results of CMR1 and CMR2 with respect to the outcome parameters IS, LVEF, CS and LS. Whereas IS, LVEF and CS improve in both groups, results in the CRP apheresis group remain significantly better in comparison to controls. The comparison reveals in detail (see also Table 1 and Table 2):For IS, CMR1 reveals an ~8% absolute or 26% relative improvement in patients with CRP apheresis compared to controls. Whereas CMR2 shows an improvement of IS in both, control and apheresis group (23.52% vs. 18.10%), the gap between both groups remains significant with a *p*-value = 0.05 in favor of the patients in the CRP apheresis group.For LVEF, CMR1 shows, on average, a 3.7% absolute or 7.8% relative better value in patients with CRP apheresis compared to controls. This initial trend, however, is not statistically significant (*p* = 0.1). Notably, CMR2 shows a 6.4% absolute or 13% relative improvement of LVEF in favor of patients in the CRP apheresis group. This improvement is statistically significant with a *p*-value = 0.03. Thus, CRP apheresis has a significant long-term effect on LVEF in patients with a CRP_grad_ > 0.6 mg/L/h.For CS, CRP apheresis shows a significant effect of 2.8 units absolute or 12% relative in CMR1 already. In CMR2, this advantage increases to 6.1 units absolute or 27% relative (*p*-value = 0.005). Hence, CRP apheresis has a significant long-term effect on CS in patients with a CRP_grad_ > 0.6 mg/L/h.For LS, there is no significant difference between the two groups in neither CMR1 nor CMR2. However, this parameter shows a significant improvement by 2.8 units in the CRP apheresis group in the period between CMR1 and CMR2, while control patients do not show a corresponding improvement.When LVEF is plotted as a function of infarct size (Figure 1), plots for CMR1 and CMR2 show a kink at infarct size ≈ 22%. That is, LVEF remains unchanged on average at the level of LVEF ≈ 55% up to about 22% of infarct size and decreases significantly beyond that point.As an aside, it should be noted that the IS of apheresis patients with a mean of 22.51% in CMR1 almost reach the kink in the curve regarding LVEF (Figure 1A) as a function of IS in CMR1 already, while controls with a mean of 30.49% are still well within the range of strongly decreasing LVEF. For CMR2, it should be noted that the mean of IS in apheresis patients is clearly left of the above-mentioned kink (Figure 1B), while control patients with a mean of 23.52% still remain in the range of decreasing LVEF.

### 3.2. Course of Myocardial Parameter Development between CMR1 and CMR2

Figure 2 illustrates the details of the development of IS, LVEF, CS and LS between CMR1 and CMR2. Notably, the variance resolution for IS (R^2^ = 0.75) is significantly larger than for the other three parameters. This may suggest that final size of IS apparently fixes first in time as compared to LVEF, CS and LS. The latter is clinically reasonable.

### 3.3. CRP_grad_ and Prognosis

#### 3.3.1. CMR1

The increase-rate of the CRP amount (CRP_grad_) during the first 12 to 32 h after onset of symptoms, proved to be a suitable measure to estimate the expected damage ca. 5 d after onset with respect to the relevant parameters IS, LVEF, LS and CS [4]. To further specify this data, we have now used a cut off value of 0.6 in CRP_grad_ in order to separate the STEMI patients in two groups. STEMI patients with CRP_grad_ < 0.6 turned out to have a “favorable” prognosis. CRP apheresis may have a small benefit in patients with a CRP_grad_ < 0.6, whereas in patients with a CRP_grad_ > 0.6, CRP apheresis is beneficial (Table 3). Indeed, in patients with a CRP_grad_ > 0.6 mg/L/h (which corresponds to approx. 2/3 of patients), CRP apheresis causes a significant improvement in the parameters IS, LVEF, and CS in CMR1.

#### 3.3.2. CMR2

The measurements of IS, LVEF, CS and LS taken in CMR2 initially give the impression that control patients would have made up for their significant disadvantage in CMR1 compared with apheresis patients after 88 d. This would call the sustainability of CRP apheresis in myocardial infarction into question. This prima facie misclassification, however, is due to two opposing effects:

First, five control patients with extremely large mean IS (which is similar in LVEF, CS and LS) of 37% (25–55%) in CMR1 but missing values in CMR2 resulted in artifactually reduced IS of control patients there by omission/missing. Secondly, three apheresis patients with very small infarct size of 14.5% in CMR1 apparently led to an increase of infarct size in the apheresis patients by omission/missing in CMR2. As shown by an analysis of the distribution of missing values in CMR2, however, this is obviously a problem of "bad data" due to missing not at random (MNAR) which could be remedied by appropriate modeling of IS, LVEF, CS and LS in CMR2 based on corresponding CMR1 values and CRP gradient of the patients concerned.

## 4. Discussion

A predictive value of the initial CRP kinetics for the extent of myocardial damage has been described several times [1,3,4,12]. The reason for these quantitative relationship between CRP and infarct size may be CRP-induced disposal of ischemic cells [7]. The loss of cardiac function in the days after STEMI is probably largely determined by the amount of CRP that surges in the first few days. Therefore, a therapeutic goal may be to lower the CRP concentration after STEMI quickly and effectively in those patients in whom a strong CRP increase occurs. The latter is possible with selective CRP apheresis. First data in humans strongly suggest that the previously damaged heart tissue can regenerate better, and the remodeling is less pronounced [4].

Wound healing starts immediately after STEMI [9] and is largely complete after a few weeks [10,11]. The initial CRP level, however, and the post-infarction architecture of the heart determines its functionality in the later life of the infarction patient [12]. According to the CMR2 data presented in this manuscript, the patients treated with CRP apheresis retain a better cardiophysiological status (Table 1). Although the patients in the control group also improve, they can no longer make up for the morphological and functional advantages that result from CRP apheresis, i.e., the differences in infarct size between the apheresis and the control group remain significant even after about 88 d (Table 1 and Table 2). In both groups, patients were also observed in whom the infarct size remained unchanged over time or even increased. Similar findings have already been made by Pokorny et al. [10]. They indicate that there could still be other influencing variables with regard to the extent of the infarction. The underlying mechanisms should be further investigated. Several other biomarkers have also been described that are prognostic for remodeling (IL-6, BNPs, troponin, MMP-9, adrenomedullin) [13,14]. Some of these are released by dying cells. The future will show whether one or the other molecule also intervenes pathologically. There is already initial clinical data announced on the blockade of adrenomedullin in sepsis, which raises hopes at least for this indication [15].

In CRP apheresis, the CRP pentamer is a direct therapeutic target for the treatment of acute inflammation. Up to the present day, CRP cannot effectively be targeted with any other anti-inflammatory therapy. It is an innovative approach that was first applied in myocardial infarction in the CAMI-1 trial [4,16]. Thus, CRP apheresis could be a valuable therapeutic option in the management of myocardial infarction patients, especially those with high CRP concentrations, defined as CRP_grad_ > 0.6.

### Limitations

Regarding the determinations of IS, LVEF, CS and LS, the following caveat should be made:

First, current CMR measurements, like any measurement, are subject to errors.

Secondly, patients had to be admitted to the clinic for CMR2 with follow-up, which explains the wide variation in time between CMR1 and CMR2 of 65 and 177 d, but complicates the interpretation of the data in terms of the remodeling process.

The CAMI-1 trial demonstrated the benefit of CRP-apheresis in acute inflammation but has some limitations. The number of patients should be enlarged and a follow-up study should only include patients with CRP_grad_ > 0.6, which may have a proper benefit of CRP removal. A prospective, multicenter, randomized, controlled trial with a larger number of patients and a follow-up period of several months is currently ongoing [“CRP Apheresis in STEMI”-trial (NCT04939805), initiated by the University of Innsbruck, Austria].

## 5. Conclusions

CRP apheresis is the first innovative therapeutic method to avoid the expansion of the cardiac infarction area and to sustainably improve cardiac performance.

## Figures and Tables

**Figure 1 jcm-11-06446-f001:**
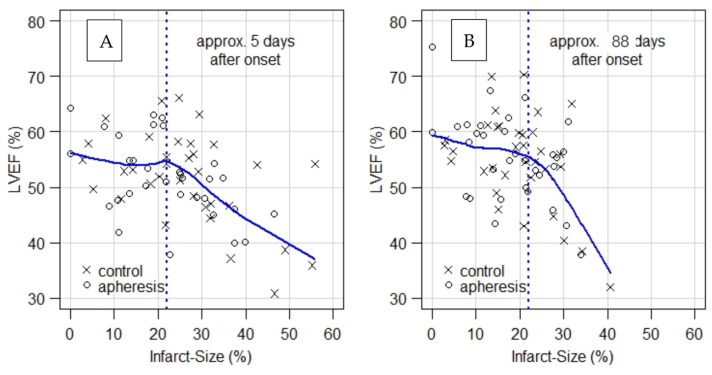
LVEF is plotted as a function of infarct size. Both plots ((**A**) with data from CMR1 and (**B**) with data from CMR2) show a kink at infarct size ≈ 22%. That is, LVEF remains unchanged on average at the level of LVEF ≈ 55% up to about 22% of infarct size and decreases significantly beyond that point. CMR1 = first Cardio Magnetic Resonance, CMR2 = second Cardio Magnetic Resonanz, LVEF = Left Ventricular Ejection Fraction.

**Figure 2 jcm-11-06446-f002:**
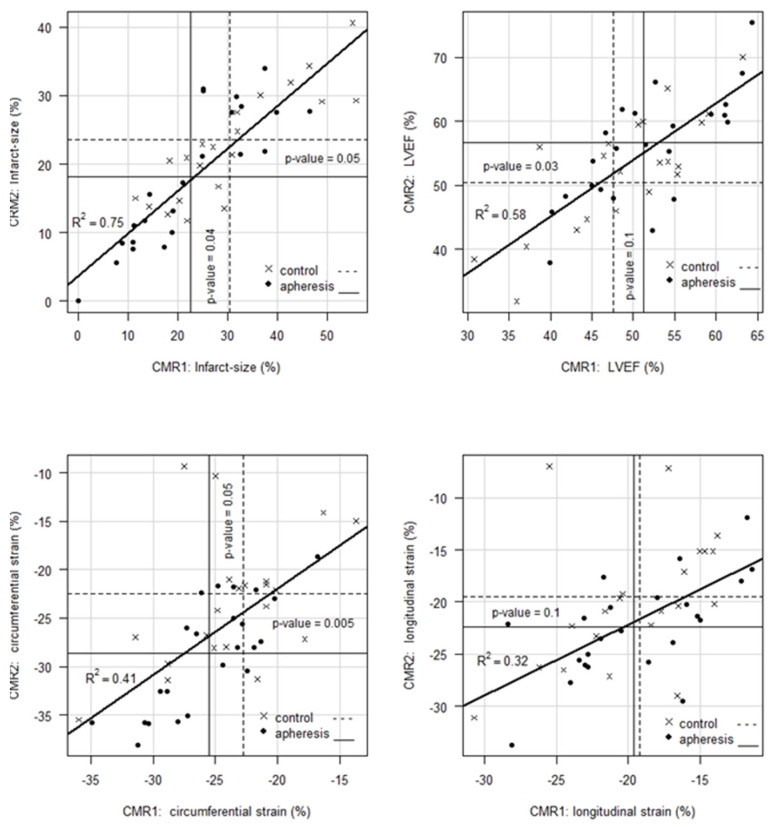
Correlation and temporal evolution of IS (%), LVEF (%), CS (%) and LS (%) in CMR1 and CMR2. The vertical/horizontal lines mark the mean values concerning CMR1 and CMR2 for control and apheresis group. The regression line plotted in each graph represents the evolution over time of both control and apheresis patients, as they are statistically identical in each case. All endpoints other than LS show significant improvements of the parameters in CMR2 compared to CMR1, especially in the apheresis group. Please see also Table 1 and Table 2.

**Table 1 jcm-11-06446-t001:** Results of CMR1 in patients with a CRP_grad_ > 0.6 mg/L/h.

Mean ± Se	CMR1			
	Control (*n* = 21/34)	Apheresis (*n* = 23/32)	*p*-Value	Improved
IS (%)	30.49 ± 2.8	22.51 ± 2.5	0.04	26%
LVEF (%)	47.58 ± 1.7	51.32 ± 1.5	0.1	7.8%
CS (%)	−22.72 ± 1.1	−25.51 ± 0.88	0.05	12%
LS (%)	−19.18 ± 1.0	−19.61 ± 1.0	0.78	2.2%

CMR1 = first Cardio Magnetic Resonance, CS = Circumferential Strain, IS = Infarc Size, LS = Longitudinal Strain, LVEF = Left Ventricular Ejection Fraction.

**Table 2 jcm-11-06446-t002:** Results of CMR2 in patients with a CRP_grad_ > 0.6 mg/L/h.

Mean ± Se	CMR2			
	Control (*n* = 21/34)	Apheresis (*n* = 23/32)	*p*-Value	Improved
IS (%)	23.52 ± 1.7	18.10 ± 2.1	0.05	23%
LVEF (%)	50.34 ± 1.7	56.73 ± 1.8	0.03	13%
CS (%)	−22.48 ± 1.3	−28.61 ± 1.2	0.005	27%
LS (%)	−19.51 ± 1.3	−22.41 ± 1.0	0.1	15%

CMR2 = second Cardio Magnetic Resonance, CS = Circumferential Strain, IS = Infarc Size, LS = Longitudinal Strain, LVEF = Left Ventricular Ejection Fraction.

**Table 3 jcm-11-06446-t003:** Dose-Dependent Effect of CRP regarding CRP gradient cut-off of 0.6 mg/L/h.

Mean ± Se	CMR1			
	Control		Apheresis	
	CRP_grad_ < 0.6 (*n* = 13)	CRP_grad_ > 0.6 (*n* = 21)	CRP_grad_ < 0.6 (*n* = 9)	CRP_grad_ > 0.6 (*n* = 23)
IS (%)	20.03 ± 3.23	30.49 ± 2.8	20.45 ± 3.30	22.51 ± 2.5
LVEF (%)	57.15 ± 1.59	47.58 ± 1.7	52.26 ± 2.22	51.32 ± 1.5
CS (%)	−27.86 ± 1.16	−22.72 ± 1.1	−24.83 ± 1.15	−25.51 ± 0.88
LS (%)	−21.73 ± 0.94	−19.18 ± 1.0	−20.1 ± 1.38	−19.61 ± 1.0

CMR1 = first Cardio Magnetic Resonance, CS = Circumferential Strain, IS = Infarc Size, LS = Longitudinal Strain, LVEF = Left Ventricular Ejection Fraction.

## Data Availability

The original contributions presented in the study are included in the article and in [4], further inquiries can be directed to the corresponding author.

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
