# Peer review of "Sustainability of C-Reactive Protein Apheresis in Acute Myocardial Infarction—Results from a Supplementary Data Analysis of the Exploratory C-Reactive Protein in Acute Myocardial Infarction-1 Study"

_jcm, 2022, doi:10.3390/jcm11216446_

Round 1
Reviewer 1 Report
This is an interesting manuscript. However, I have some comments/suggestions to the authors.
Since the numbers of patients are small, I would suggest to the authors to present the power analysis.
There is no need to put some results in the 2.1. Statistical analysis section (e.g. "In CAMI-1, about 2/3 of patients had a CRP-Gradient >0.6 102 mg/L/h and significant myocardial damage.
These patients also had a significant benefit from CRP apheresis."
In Discussion the authors should at least hypothesize about possible pathophysiological mechanisms which could explain why CRP apheresis might benefit AMI patients.
Author Response
This is an interesting manuscript. However, I have some comments/suggestions to the authors.
Since the numbers of patients are small, I would suggest to the authors to present the power analysis.
To calculate the efficacy of the sustained effect of CRP apheresis in this manuscript, results from a supplemental data analysis of the exploratory CAMI-1 trial Table 1 of the previous publication “C-reactive protein apheresis as an anti-inflammatory therapy in acute myocardial infarction: results of the CAMI-1 study” were used. The results of this power analysis are in the table below:
Parameter |
ntreat / ncontrol |
µtreat – control |
SD |
Power |
IS |
23 / 21 |
- 5.42 |
6.1 |
0.66 |
LVEF |
23 / 21 |
6.41 |
5.9 |
0.82 |
CS |
23 / 21 |
- 6.13 |
4.8 |
0.92 |
LS |
23 / 21 |
- 2.91 |
4.5 |
0.32 |
There is no need to put some results in the 2.1. Statistical analysis section (e.g., "In CAMI-1, about 2/3 of patients had a CRP-Gradient >0.6 102 mg/L/h and significant myocardial damage. These patients also had a significant benefit from CRP apheresis."
We removed this passage.
In Discussion the authors should at least hypothesize about possible pathophysiological mechanisms which could explain why CRP apheresis might benefit AMI patients.
In the introduction between line 63 and line 78 we explained the pathophysiology. We have discussed additional aspects in the discussion in line 234 - 250.
Reviewer 2 Report
Thank you for the opportunity to review an interesting paper on the role of CRP apheresis in acute myocardial infarction. The concept is novel, and the study's results are clearly presented.
1. Please comment on the lack of differences in the baseline characteristics. Please refer to the main paper or reduplicate the table with baseline characteristics as a supplementary table.
2. The use of bullet points in Results and Discussion seems strange.
3. Please comment on feasibility of CRP apheresis in AMI patients.
4. The Discussion is very short and focuses, in fact, only on CRP. Please discuss the predictive role of other biomarkers in assessing post-MI remodeling and the possible interplay between them and CRP.
· Refer to Adv Interv Cardiol. 2021 Mar;17(1):21-32. doi: 10.5114/aic.2021.104764.
· Pol Arch Intern Med. 2022 Feb 28;132(2):16150. doi: 10.20452/pamw.16150.
Minor:
1. Please avoid abbreviations in the title.
2. Some typos should be corrected - "relativ" should be "relative".
3. In Figure 1, figures are marked as 5 and 6. Please clarify / correct.
Author Response
Thank you for the opportunity to review an interesting paper on the role of CRP apheresis in acute myocardial infarction. The concept is novel, and the study's results are clearly presented.
We thank the reviewer for his kind and supportive comments.
- Please comment on the lack of differences in the baseline characteristics. Please refer to the main paper or reduplicate the table with baseline characteristics as a supplementary table.
We commented with reference to the original publication (line 100-109).
- The use of bullet points in Results and Discussion seems strange.
We hope that the bullet points will contribute to clarity.
- Please comment on feasibility of CRP apheresis in AMI patients.
We included to comment on the feasibility of CRP apheresis now (line 56-57).
- The Discussion is very short and focuses, in fact, only on CRP. Please discuss the predictive role of other biomarkers in assessing post-MI remodelling and the possible interplay between them and CRP.
We discussed this important point and included the literature (line 245-250).
- Refer to Adv Interv Cardiol. 2021 Mar;17(1):21-32. doi: 10.5114/aic.2021.104764.
- Pol Arch Intern Med. 2022 Feb 28;132(2):16150. doi: 10.20452/pamw.16150.
Minor:
- Please avoid abbreviations in the title.
We have changed the title accordingly. Thank you very much for this advice!
- Some typos should be corrected - "relativ" should be "relative".
Done throughout the manuscript.
- In Figure 1, figures are marked as 5 and 6. Please clarify / correct.
We corrected this mistake which was obviously introduced by some unforeseen formatting of WORD.
Round 2
Reviewer 1 Report
The authors have answers to all my comments/suggestions so I do not have any further comments.